# Ocular Complications Following Vaccination for COVID-19: A One-Year Retrospective

**DOI:** 10.3390/vaccines10020342

**Published:** 2022-02-21

**Authors:** Abid A. Haseeb, Omar Solyman, Mokhtar M. Abushanab, Ahmed S. Abo Obaia, Abdelrahman M. Elhusseiny

**Affiliations:** 1Department of Ophthalmology and Visual Sciences, University of Illinois at Chicago, Chicago, IL 60612, USA; haseeb2@uic.edu; 2Department of Ophthalmology, Qassim University Medical City, Qassim University, Buraydah 52571, Saudi Arabia; o.solyman@rio.edu.eg; 3Department of Ophthalmology, Research Institute of Ophthalmology, Giza 11261, Egypt; mokh211@yahoo.com (M.M.A.); a.sobh@rio.edu.eg (A.S.A.O.); 4Department of Ophthalmology, Harvey and Bernice Jones Eye Institute, University of Arkansas for Medical Sciences, Little Rock, AR 72205, USA

**Keywords:** acute macular neuroretinopathy, corneal graft rejection, coronavirus, COVID-19, SARS-CoV-2, uveitis, vaccination

## Abstract

Vaccination efforts as a mitigation strategy in the corona virus disease 2019 (COVID-19) pandemic are fully underway. A vital component of understanding the optimal clinical use of these vaccines is a thorough investigation of adverse events following vaccination. To date, some limited reports and reviews have discussed ocular adverse events following COVID-19 vaccination, but a systematic review detailing these reports with manifestations and clinical courses as well as proposed mechanisms has yet to be published. This comprehensive review one-year into vaccination efforts against COVID-19 is meant to furnish sound understanding for ophthalmologists and primary care physicians based on the existing body of clinical data. We discuss manifestations categorized into one of the following: eyelid, orbit, uveitis, retina, vascular, neuro-ophthalmology, ocular motility disorders, and other.

## 1. Introduction

Since the time that the first vaccines against the severe acute respiratory failure coronavirus 2 (SARS-CoV-2) and ensuing corona virus disease2019 (COVID-19) were approved for emergency authorization use by the Food and Drug Administration (FDA) in late 2020, an enormous amount of speculation has surrounded the discourse around vaccination and COVID-19 [1,2,3,4]. It remains the consensus opinion in clinical practice—and the opinion of the authors—that vaccination and subsequent booster administration against COVID-19 is a vital epidemiologic factor in mitigating the devastating effects of the COVID-19 pandemic. It is nevertheless essential that physicians and researchers investigate the possible adverse outcomes due to vaccination against COVID-19.

Previous research has demonstrated a link between COVID-19 infection and ocular complications, direct or indirect [5,6,7,8,9,10,11,12,13]. It has been well-documented that conjunctivitis, scleritis, orbital inflammatory disease, phlyctenular keratoconjunctivitis and retinal involvement may take place in COVID-19 infection. It is thus vital to also investigate the relationship between COVID-19 vaccination and ocular complications. A considerable number of reports and retrospective case studies have reported on possible adverse effects of vaccination against COVID-19 approximately one year into the dissemination of these vaccines [14,15,16]. In this review, we seek to provide a rigorous description of these findings based on a comprehensive review and statistical analysis of the literature.

## 2. Materials and Methods

We performed a PubMed search for articles of interest using search terms beginning with “coronavirus vaccine” or “COVID vaccine” followed by “ocular”, “palsy”, “cornea”, ”rejection”, “uveitis”, “optic neuritis”, “optic neuropathy” and “retina”. Articles were included if they were case reports or retrospective studies describing adverse ocular manifestations following any vaccination against COVID-19 between December 2020 and December 2021.

We thereafter characterized articles as belonging to one of the following categories of adverse event: eyelid, orbital, corneal, uveitis, retinal, vascular, neuro-ophthalmological, ocular motility disorders and unspecified. When possible, statistical analysis was performed for each category regarding age, sex, visual acuity, and any other pathology-specific characteristics of the categories in question. Continuous variables were reported as mean ± one standard deviation.

## 3. Adverse Ocular Events: Patient Overview

In total, 58 articles were included in our review. These findings are detailed in Table 1. Of these 58 studies, 28 (48.3%) were case reports, 5 (8.6%) were case series, 22 (37.9%) were letters to the editor, and 3 (5.2%) were photo essays. A total of 94 patients were included. Of 90 patients with documented age information, the mean age at the time of presentation was 46.9 ± 18.4 years. Of 91 patients with documented gender, there were 50 (54.9%) females and 41 (45.1%) males. Of the 87 cases in which vaccine information were present, BNT162b2 mRNA SARS-CoV-2 (BioNTech/Pfizer, Mainz, Germany) was reported 55 (63.2%) times, AZD1222 ChAdO×1 nCoV-19 (AstraZeneca, Cambridge, UK, also marketed as the CoviShield Serum Institute of India vaccine) was reported 20 (22.9%) times, Moderna COVID-19 Vaccine (ModernaTX, Inc., Cambridge, MA, USA) was reported 6 (6.9%) times, BBIBP-CorV (Sinopharm, Beijing, China) was reported 3 (3.4%) times, Corona Vac (Sinovac Biotech Ltd., Beijing, China) was reported 2 (2.3%) times, and Gam-COVID-Vac/Sputnik V (Gamaleya Institute, Moscow, Russia) was reported once (1.1%). Vaccine ordinal dose was reported 81 times; 45 (55.6%) cases were after the first dose, 35 (43.2%) were after the second dose, and one (1.2%) was after a 3rd (booster shot) dose.

In our literature review, we found numerous ophthalmic adverse events following COVID-19 vaccination. Because some phenomena were reported several times and others were reported only once, we primarily discuss mechanisms and clinical considerations for phenomena which have occurred several times. Nevertheless, it is important to note that an important limitation of this analysis is that it is a retrospective review and not a cohort study. Despite the fact that we discuss mechanisms, in the absence of definitive underlying pathophysiologic processes, we must recognize the possibility that some adverse events—particularly those which are especially rare—are due to random chance.

## 4. Eyelid

In general, reports of eyelid manifestations following vaccination against COVID-19 are limited. Of the 2 reports (6 patients) that we reviewed, 5 of 6 patients (83.3%) were females and the mean age at the time of presentation was 48.7 ± 13.5 years [17,18]. Visual acuities were not reported.

In one study, Austria et al. reported on a series of three women who each presented with unilateral edema more prominent in the upper eyelid following vaccination with the BNT162b2 vaccine [17]. All three patients were middle-aged women (aged 32, 43, and 43) and they were all treated differently. One patient was treated with observation, another with antihistamines, and one with oral steroids. All patients had complete resolution of orbital edema within two days.

Elsewhere, Mazzatenta et al. described a case series of three patients who developed ecchymotic or purpuric lesions on the upper eyelids 1 to 3 weeks following vaccination with the BNT162b2 vaccine [18]. In all three cases, lesions were bilateral and resolved within approximately two weeks.

Regarding the mechanism of these findings, Austria et al. proposed in their report that eyelid changes may be mediated by complement activation which increased complement mediators within the tear duct via leakage of plasma [17,18]. Further investigation is required to support this hypothesis.

## 5. Orbit

A total of three reports (3 patients, 4 eyes) commented on orbital manifestations following vaccination against COVID-19. These cases are described below.

### 5.1. Superior Ophthalmic Vein Thrombosis

Two cases commented on superior ophthalmic vein thrombosis. Bayas et al. reported a case of a 55-year-old woman who presented with conjunctival injection, retro-orbital pain, and diplopia seven days after getting vaccination with the AZD1222 vaccine [19]. Magnetic resonance imaging (MRI) of brain and orbit with contrast showed superior ophthalmic vein thrombosis with no contrast filling and bilateral T2 signal intensity of the superior ophthalmic vein. Laboratory values revealed secondary immune thrombocytopenia. The patient later developed a transient right-sided hemiparesis and aphasia, and MRI testing demonstrated a left parietal lobe ischemic stroke. Ultimately the patient was treated with anticoagulation and discharged. Elsewhere, Panovska-Stavridis et al. reported on a case of a 29-year-old woman (Figure 1) who developed left orbital swelling, severe headache, and blurred left eye vision 10 days after receiving the AZD1222 vaccine [20]. MRI imaging demonstrated central filling defects and a diagnosis of superior ophthalmic vein thrombosis was made. The patient was treated with intravenous (IV) immunoglobulins 1 g/kg for two days, followed by an oral prednisolone taper. Concurrently, the patient was placed on rivaroxaban 15 mg twice daily for 21 days as well as broad-spectrum antibiotics. The authors reported excellent response within 5 days and the patient’s thrombocytopenia also resolved.

### 5.2. Tolosa-Hunt Syndrome

Chuang et al. reported on a case of a 45-year-old male who developed left eye pain with progressive ptosis, decreased vision, and binocular diplopia seven days after receiving an unspecified COVID-19 vaccine [21]. The patient had an afferent pupillary defect (APD) and complete ophthalmoplegia. Imaging with computed tomography (CT) and MRI of the brain were most consistent with cavernous sinus thrombosis. In the setting of this constellation of findings, the patient was diagnosed with Tolosa–Hunt syndrome.

Diagnostic criteria for Tolosa–Hunt syndrome require unilateral orbital affection with associated paresis of one or more of the 3rd, 4th, and 6th cranial nerves [22]. Cavernous sinus thrombosis with Tolosa–Hunt syndrome has been reported sparingly, but a previous report of it following hepatitis-B vaccination has been described [23]. On our review of other vascular phenomena, we found limited reports of central retinal vein occlusion (CRVO), branch retinal vein occlusion (BRVO), and hemi-retinal vein occlusion (HRVO) (discussed later).

### 5.3. Mechanisms

The mechanisms underlying a possible hypercoagulable state following vaccination have not yet been completely elucidated. However, Schultz et al. previously reported five cases of severe venous thromboembolism—four of which were cerebral venous thrombosis—following vaccination against COVID-19 [24]. Because all cases resolved with transfusions and had an absence of hemolysis, the authors ruled out thrombotic thrombocytopenic purpura and immune thrombocytopenia. However, in all cases, there was a high level of antibodies to platelet factor-4 (PF4)-polyanion complexes, suggesting a vaccine-related variant of the phenomenon of heparin-induced thrombocytopenia termed vaccine-induced thrombotic thrombocytopenia (VITT) [24]. Indeed, thrombosis and thrombocytopenia has previously been reported following the use of Measles-Mumps-Rubella [25,26,27,28], influenza [29], pneumococcal [30], smallpox [31], and COVID-19 vaccines [24,32,33,34,35], but it is unclear if these would all fall into the category of VITT. Thrombotic microangiopathies have previously been reported following influenza vaccination and linked to thrombotic thrombocytopenic purpura, but these reports are rare [36].

It is our clinical recommendation that any patient presenting with thrombosis—with or without ophthalmic manifestations—should be tested for thrombocytopenia, response to platelet transfusion, and the presence of anti-PF4 complex antibodies.

## 6. Uveitis

Given the strong association between uveitis and immunologic phenomena, it would be expected that there is some relationship between vaccination against COVID-19 and uveitis. Of the 14 reports we reviewed dealing with uveitis after COVID-19 vaccination, 34 patients (44 eyes) were reported on. Of these 34, 19 (55.9%) were males, 15 (44.1%) were females, and the average age at the time of presentation was 47.6 ± 16.3 years. For the 34 patients, average time from vaccination to development of ophthalmic symptoms was 8.0 ± 8.6 days. Ten patients (29.4%) presented with bilateral manifestations. For the 40 eyes which had presenting visual acuity information, the mean presenting visual acuity was logMAR 0.421 ± 0.455 (20/52 in Snellen notation). For the 35 eyes which had both presenting and final visual acuity at last follow-up, these values were 0.434 ± 0.426 (20/54 in Snellen notation) and 0.085 ± 0.166 (20/24 in Snellen notation), respectively (*p* < 0.001).

### 6.1. Uveitis Flares

Previously, we described a case of an 18-year-old girl with a history of antinuclear antibody positive oligoarticular juvenile idiopathic arthritis (JIA) (but no prior history of uveitis) who presented with bilateral anterior uveitis 5 days after the second dose of the Sinopharm COVID-19 vaccine [37]. Examination was notable for anterior uveitis, and optical coherence tomography (OCT) showed hyperreflective dots and circulating cells in the anterior chamber (AC). Uveitis in both eyes resolved gradually after topical steroid treatment without recurrence. Similar to our case, Jain and Kalamkar reported on a 27-year-old man with past medical history of JIA and one previous episode of uveitis who developed a uveitis flare-up in the left eye (OS) two days after receiving the AZD1222 vaccine [38]. Similar to our previous report, the patient demonstrated resolution with topical steroids and cycloplegic drops.

Numerous other reports have been made. Mudie et al. described a case of a 43-year-old woman who presented with eye pain, redness, and photophobia bilaterally 3 days after her second dose of the BNT162b2 [39]. Examination was notable for a thickened choroid and pronounced inflammation in the AC and the vitreous cavity. The patient responded well to oral and topical corticosteroids with a mild recurrence after the initial attempt to taper these drugs. Renisi et al. described a similar case in a 23-year-old man who developed pain, photophobia, and a red eye four days after receiving the second dose of the BNT162b2 vaccine [40]. Examination revealed conjunctival hyperemia, posterior synechiae, and AC cells with keratic precipitates (KP) in the lower quadrants. The patient demonstrated initial improvement on topical dexamethasone and atropine drops daily over 3 weeks, then demonstrated complete resolution at 6 weeks.

Ishay et al. reported a case of a 28-year-old male with past medical history of Behçet’s disease on colchicine twice daily [41]. Ten days after receiving the BNT162b2 vaccine, the patient developed left eye pain, redness, and blurred vision. Examination revealed severe panuveitis. Unlike the previous cases, the patient was successfully treated with five days of pulse-dose IV methylprednisolone followed by oral (PO) corticosteroids and azathioprine.

In a different case, Herbort and Papasavvas reported on a 53-year-old male with pre-existing herpes keratouveitis which was inactive for 18 months without treatment [42]. Five days after receiving the Moderna COVID-19 vaccine, the patient presented with a severe flare-up of disease, including numerous KPs and elevated intraocular pressure to 41 mmHg. The patient was treated with PO valacyclovir 500 mg, topical dexamethasone, dorzolamide, and timolol. Over 6 days of treatment, the patient demonstrated an improvement in flare, and KPs resolved almost completely after 3 weeks.

To date, the largest and only multicenter study investigating a relationship between uveitis and COVID-19 vaccination was conducted by Rabinovitch et al. [43]. In their study, the authors examined 23 eyes of 21 patients (mean age of 51.3 years) who developed uveitis after vaccination against COVID-19 with the BNT162b2 vaccine. These patients presented with uveitis an average of 7.5 ± 7.3 days after vaccination. A total of 8 of the 21 patients had pre-existing uveitis, though average time since last flare was one year, and no patients had recent changes in medication regimen. Eight of 21 patients presented after first dose of vaccination and 13 of 21 presented after second dose of vaccination. Six of the 21 patients had pre-existing uveitis-related diseases, including ankylosing spondylitis, psoriasis, Crohn’s disease, and herpes zoster (VZV) ophthalmicus. Two patients had bilateral disease presentation. Twenty-one of 23 eyes had anterior uveitis and two eyes had multiple evanescent white dot syndrome. Nineteen of 21 patients were treated with steroids, most commonly prednisone or dexamethasone, and all 19 of these patients demonstrated complete resolution of inflammation. Two of 21 patients did not undergo treatment but demonstrated significant improvement, nevertheless.

### 6.2. Choroiditis

Two reports have been made connecting choroiditis with vaccination against COVID-19. Goyal et al. reported on a 34-year-old male who developed ocular pain and nasal redness OS as well as a floater in the right eye (OD) progressing to severe vision loss 4 days after receiving the AZD1222 vaccine [44]. At presentation, his visual acuity was 20/120 OD and 20/20 OS. The patient’s fundus exam demonstrated multiple bilateral oval lesions at the level of the choroid with serous detachment, consistent with a diagnosis of bilateral multifocal choroiditis. The patient was treated with a PO prednisolone taper to beginning at 100 mg daily and demonstrated significant improvement in inflammation and subretinal fluid after 11 days of treatment. His visual acuity at the last follow-up was 20/20 in both eyes.

Another report by Pan et al. described a 50-year-old woman who developed bilateral blurred vision her 5 days after receiving an unspecified Vero cell-based vaccine in China [45]. Her examination revealed a pale, blurry optic disc, absent foveal reflex, and macular edema. Imaging with fluorescein angiography was consistent with bilateral choroiditis. The patient’s vision and inflammation improved considerably over 5 weeks with periocular triamcinolone acetamide and PO prednisone.

### 6.3. Vogt–Koyanagi–Harada Disease

Vogt–Koyanagi–Harada (VKH) disease is a T-lymphocyte mediated multi-system disease affecting the auditory system, skin, meninges, and eye [46,47]. Ophthalmologically, it causes a granulomatous panuveitis often affecting young adults, and may also present with exudative retinal detachments and a sunset glow fundus [46,47,48,49]. As it is an autoimmune disease resulting from antibodies against melanocytes-associated antigens, the robust immune response mounted by patients following vaccination against COVID-19 may be of importance to patients living with VKH or other autoimmune diseases.

Papasavvas and Herbort reported on a 43-year-old woman who had a previous history of VKH disease which was under control for 6 years using mycophenolate, cyclosporine, and intermittent infliximab infusions [50]. However, six weeks after the second dose of the BNT162b2 vaccine, the patient presented with severe reactivation of disease. Although her visual acuity remained 20/20 OD and OS, she had severe AC inflammation with 3–4 small mutton-fat KPs as well as bilateral exudative retinal detachments. Several hypofluorescent dark dots were present on indocyanine green angiography (ICGA), which was also observed when the patient was first diagnosed with VKH disease. This flare was ultimately controlled using infliximab. Furthermore, the authors speculated that the flare had occurred with the second dose because the patient’s last infliximab infusion had been performed 3.5 weeks before her first dose of the vaccine but 7.5 weeks before the second dose. This case in particular highlights the possibility that COVID-19 vaccination may be associated with reactivation or exacerbation of pre-existing autoimmune disease.

In another report, Saraceno et al. described a 62-year-old female who developed acute bilateral loss of vision two days after receiving the AZD1222 vaccine [51]. She was found to have visual acuity of 20/600 OD and 20/200 OS. On examination, she had 2+ AC cells and 1+ vitreous cells bilaterally. Fundus examination revealed serous retinal detachments and optic disc hyperemia bilaterally (Figure 2). OCT demonstrated subretinal hyperreflective dots. In this case also a diagnosis of VKH was made. The patient was treated with PO prednisone. Intravenous therapy was avoided due to a restriction in available hospital beds. Within four days, the patient’s visual acuity improved to 20/60 OD and 20/80 OS. At a three week follow up, the patient remarkably demonstrated visual acuity of 20/20 in both eyes with no signs of inflammatory activity and resolution of the exudative retinal detachments. Of note, the authors also described a case of VKH which developed in a 37-year-old female patient two weeks after she tested positive for COVID-19 on a polymerase chain reaction (PCR) test—i.e., in the setting of COVID-19 infection—as opposed to after vaccination [51]. Similarly, the patient had vision loss and signs of inflammation: She had KPs OD, mild vitritis bilaterally, and fluorescein angiography demonstrated bilateral optic disc hyperfluorescence due to leakage. The patient had bilateral serous retinal detachments as well. This patient was also treated with PO prednisone with improvement in vision and resolution of retinal detachments. These reports of VKH in the setting of both COVID-19 vaccination and infection suggest that there may be a common immunologic link between vaccination against and infection with COVID-19 which connects them both to the development of VKH.

### 6.4. Acute Retinal Necrosis

Finally, Mishra et al. reported on a 71-year-old man who developed reactivation of VZV OD following his first dose of the AZD1222 vaccine [52]. The patient presented with panuveitis, circumcorneal congestion, multiple fine KPs, vitritis, and widespread acute retinal necrosis. He was successfully treated with 12 weeks of PO valacyclovir 1 g three times daily and a PO prednisolone taper starting at 40 mg.

### 6.5. Acute Zonal Occult Outer Retinopathy

Maleki et al. reported on a 33-year-old woman who developed bilateral photopsias and a progressive nasal field defect OS 10 days after receiving the second dose of the Moderna COVID-19 vaccine [53]. Imaging with OCT was demonstrative of an outer layer segmental disruption OS. A diagnosis of acute zonal occult outer retinopathy (AZOOR) was made.

### 6.6. Mechanisms

The primary uveitic phenomenon we encountered on our review was new uveitis or flare-ups of pre-existing disease. Uveitis has previously been documented following numerous vaccines, most commonly the Bacille Calmette–Guerin, hepatitis B, human papillomavirus, influenza, measles-mumps-rubella, and varicella vaccines [54,55,56,57,58]. One review of 276 found that 199 (72.1%) cases were in women [58].

There are several possible mechanisms underpinning the development of post-vaccination uveitis. Fraunfelder et al. previously studied the connection between the hepatitis B vaccine and uveitis and proposed that a delayed-type hypersensitivity reaction and immune complex deposition following vaccination leads to uveitis [56]. The authors also proposed that adjuvants play a role in this immunologic process, though this does not apply to COVID-19 vaccination. Elsewhere, Aguirre et al. reported a uveitic reaction in dogs following vaccination with canine adenovirus 1, which was found to be a type III hypersensitivity reaction involving antigen-antibody complexes present in the aqueous humor [59]. Given the previously established fact that SARS-CoV-2 RNA has been found in human aqueous humor and other ocular tissues, a similar inflammatory reaction involving immune complex deposition is likely [60,61].

In their major review of uveitis following COVID-19 vaccination with the BNT162b2 vaccine, Rabinovitch et al. proposed that the possible causal mechanism is vaccine-induced type I interferon secretion [43]. The authors proposed that the vaccine mRNA activates RNA-sensing molecules including TLR3, TLR7, MDA5, and RIG-I which drive autoimmune processes in these patients. While not mutually exclusive, we favor the phenomenon of immune-complex deposition as the primary driver of COVID-19 vaccine-related uveitis. However, it has been reported that COVID-19 vaccines use the modified nucleobase N1-methylpseudouridine in order to dampen immunostimulatory potential [62]. Further investigation is required to evaluate the extent to which this impacts COVID-19 vaccine-related uveitis.

## 7. Cornea

A number of studies have reported on adverse events at the level of the ocular surface following vaccination to COVID-19. On our review, 11 reports (15 patients, 18 eyes) described corneal manifestations following vaccination against COVID-19. Of these 15 patients, 9 (66.7%) were female and 6 (33.3%) were male. The mean age at the time of presentation was 61.33 ± 15.5 years, and the average time from vaccination to development of ophthalmic symptoms was 11.8 ± 6.2 days. Three patients (20.0%) presented with bilateral involvement. For the 17 affected eyes which had reported visual acuity, the mean visual acuity was logMAR 1.09 ± 0.858 (20/247 in Snellen notation) at presentation. Only 5 studies (6 patients, 7 eyes) reported on baseline, post-transplantation visual acuity of patients who underwent graft rejection after vaccination. For these patients, baseline visual acuity was logMAR 0.204 ± 0.309 (20/32 in Snellen notation), whereas visual acuity at presentation after vaccination was logMAR 0.871 ± 0.694 (20/149 in Snellen notation). This difference was significant (*p* = 0.007). For the 13 eyes which had both presenting and final visual acuities reported, the mean visual acuities were 1.215 ± 0.878 (20/328 in Snellen notation) and 0.482 ± 0.793 (20/61 in Snellen notation), respectively (*p* < 0.001).

### 7.1. Graft Rejection

Several reports have described complications involving corneal transplant rejection following vaccination to COVID-19. Phylactou et al. reported on a pair of cases [63]. First, they described a 66-year-old woman with Fuchs endothelial corneal dystrophy (FECD) status-post unilateral Descemet’s membrane endothelial keratoplasty (DMEK) transplant in the right eye who received the BNT162b2 vaccine 14 days after DMEK. Seven days after receiving her vaccination, she presented with a visual acuity of 20/120 OD. Examination revealed moderate conjunctival injection, diffuse corneal edema, fine KPs, and 1+ AC cells. Central corneal thickness (CCT) was 652 µm, significantly increased from 525 µm one week after transplantation. She was diagnosed with acute unilateral graft rejection. She was treated with an increase in frequency of topical steroids and one week later demonstrated 20/20 vision OD with a clear cornea and decreased inflammation. The authors also reported on an 83-year-old woman with bilateral DMEK transplants for FECD 3 and 6 years before developing acute bilateral endothelial rejection, 3 weeks after her second dose of the BNT162b2 vaccine. In this case also, the patient was treated with topical steroid drops and demonstrated significant improvement at a one-week follow-up.

Several other reports have also been made. Crnej et al. reported on a 71-year-old patient who underwent DMEK surgery 5 months earlier and developed acute unilateral graft rejection 7 days after receiving his second dose of the BNT162b2 vaccination [64]. In that case, the patient was treated successfully with topical dexamethasone 1 mg/mL every two hours.

Wasser et al. reported on a pair of men, aged 56 and 73, both with a history of penetrating keratoplasty (PKP) due to keratoconus, who developed acute corneal graft rejection 2 weeks after receiving their first dose of the BNT162b2 vaccine [65]. Similar to other cases of graft rejection, both patients presented with vision loss, corneal edema, and KPs. Of note, the 56-year-old man had pre-existing grafts in both eyes but only his right eye (PKP done 25 years earlier) had rejection, while his left eye (PKP done 7 years earlier) remained intact. Both patients were successfully treated with hourly dexamethasone and oral prednisone 60 mg/day. Rallis et al. reported on a similar case of a 68-year-old woman with previous bilateral lamellar Descemet stripping automated endothelial keratoplasty (DSAEK) for previous FECD and a left re-do PKP for failed DSAEK [66]. She presented with pain and redness and rapid vision loss OS four days after receiving her first dose of the BNT162b2 vaccine. Her examination demonstrated corneal punctuate straining, corneal graft edema, Descemet’s folds, and scattered KPs, all in the left eye only (Figure 3). Similar to the case mentioned in Wasser et al., her pre-existing right eye graft was completely unaffected. At a three-week follow-up, she demonstrated complete resolution of these symptoms following hourly topical dexamethasone 0.1% and a week of PO acyclovir 400 mg five times daily to cover for herpes simplex keratitis.

In comparison, Abousy et al. described a case of a 73-year-old woman with previous bilateral DSEK for FECD who presented with bilateral decreased vision, ocular pain, and photophobia four days after her second dose of the BNT162b2 vaccine [67]. Examination revealed decreased vision to 20/200 OD and 20/40 OS as well as corneal edema OD. Further examination revealed Descemet folds bilaterally. The patient was diagnosed with bilateral graft rejection. The patient was initiated on topical prednisolone acetate 1% four times per day. She had initially persistent and worsening symptoms on this regimen at her 28-day follow-up, so prednisolone frequency was increased to hourly, and Muro ointment was added at bedtime. Prednisolone was tapered with improvement and at a two-month follow-up, the patient’s vision had improved to 20/50 OD and 20/25 OS, with significantly decreased corneal edema bilaterally., Taken together, these cases suggest that graft rejection can be unilateral or bilateral post-COVID-19 vaccination.

Similar cases of post-PKP were reported separately by Ravichandran et al., Nioi et al., and Parmar et al. in adult patients [68,69,70]. Nioi et al. uniquely found that their 44-year-old female patient had severe vitamin D deficiency concurrently with rejection, so the patient was treated with topical dexamethasone and vitamin D supplementation. After initial resolution at four weeks, she again had an episode of rejection concurrently with persistent vitamin D deficiency, and steroid drops were re-started with higher doses of vitamin D, which resulted in sustained resolution. Vitamin D deficiency has previously been demonstrated to play a vital role in adverse effects following solid organ transplantation, namely allograft rejection [71,72,73]. It plays a vital role the expression of IL-2 and interferon-mRNA, downregulates T-cell mediated cytotoxicity, and suppresses major histocompatibility complexes of immunomodulators of dendritic cells [68,74,75]. The authors supported this explanation for their patient.

### 7.2. Corneal Melting

Khan et al. reported on a 48-year-old man who developed profound vision loss to light perception three weeks after receiving his first dose of the AZD1222 vaccine [76]. He was found to have diffuse conjunctival and ciliary congestion, corneal melting and perforation with diffuse corneal haze, uveal tissue prolapse, and bilateral massive choroidal detachment on B-scan ultrasonography.

### 7.3. Mechanisms

Our review primarily revealed several cases of corneal graft rejection, both unilateral and bilateral. Corneal transplant rejection has previously been reported—albeit rarely—following influenza, hepatitis B, tetanus, and yellow fever vaccinations [77,78,79,80]. In the setting of this pre-existing precedent, it is not surprising that the highly immunogenic vaccines to COVID-19 present similar risks.

There are several possible mechanisms underlying corneal graft compromise following vaccination. One hypothesis proposed by Steinemann et al. asserts that elevated vascular permeability following vaccination compromises the native-state immunologic privilege possessed by the cornea [77]. This theory is supported by the finding of graft edema, as demonstrated on our review. In the same case series, the Steinemann et al. also proposed that immunization may induce expression of the major histocompatibility complex (MHC) of the cornea, as various organ grafts result in enhancement of MHC antigenic expression after rejection [77]. Donor cells with no MHC expression are thereafter targeted by the host immune cells due to poor immunogenicity [77,81].

Another mechanism for corneal graft rejection proposed in the setting of vaccination to COVID-19 by Abousy et al. revolves around the finding that SARS-CoV-2 RNA is present in the aqueous humor of patients with asymptomatic infections [61,67]. Likewise, Sawant et al. found that SARS-CoV-2 RNA was found in the corneas of postmortem COVID-19 patients [60]. In the setting of vaccination for COVID-19 during ongoing or previous asymptomatic infection, then, it is possible that antibody-antigen complexes would be formed in large quantities with the subsequent development of profound inflammation, again compromising the integrity of corneal grafts.

Regardless of the mechanism, we propose that ophthalmologists consider examining patients with corneal grafts prior to vaccination against COVID-19 in order to evaluate underlying inflammatory processes which may be further exacerbated by the introduction of a profound immunogenic stimulus such as a COVID-19 vaccine. Particularly if we consider the effect of different immune processes to be additive, it may be optimal for patients with corneal grafts to delay COVID-19 vaccination if experiencing a transient inflammatory process around the time of vaccination. However, given the fact that the cases of corneal graft rejection have been successfully managed with topical steroids, whereas COVID-19 infection presents grave individual and epidemiologic risks, we do not recommend that patients avoid receiving the vaccine altogether. 

## 8. Retina

Preliminary reports suggest that retinal adverse events are possible following vaccination against COVID-19. On our review, 12 reports (14 patients, 19 eyes) commented on retinal manifestations following vaccination against COVID-19. One report did not include information on the sex of two patients. Of the remaining 12 patients, 10 (83.3%) were women and 2 (16.7%) were men. The mean age at the time of presentation was 24.8 ± 4.8 years, and the average time from vaccination to development of ophthalmic symptoms was 3.1 ± 2.4 days. Five patients (35.7%) presented with bilateral involvement. For the 19 affected eyes which had presenting visual acuities, the mean visual acuity was logMAR 0.138 ± 0.325 (20/27 in Snellen notation) at presentation. For the 6 eyes which had both presenting and final visual acuities reported, the mean visual acuities were 0.350 ± 0.465 (20/45 in Snellen notation) and 0.030 ± 0.067 (20/21 in Snellen notation), respectively (*p* = 0.138).

### 8.1. Acute Macular Neuroretinopathy

Acute macular neuroretinopathy (AMN) is a rare disease, commonly affecting adult females, which frequently presents with the acute onset of paracentral scotomas affecting one or both eyes [82,83,84,85]. Fundus exam may demonstrate reddish-brown petaloid perifoveal lesions with the tip pointed toward the fovea [83,86]. There are no known treatment modalities for AMN, and vision changes may be permanent [85].

A number of reports suggest that there is an association between vaccination for COVID-19 and development of AMN. Bøhler et al. reported on a 27-year-old female with no past medical history who developed flu-like symptoms followed by a paracentral scotoma OS two days after receiving the AZD1222 vaccine [87]. Examination was notable only for a teardrop-shaped lesion nasal to the fovea. Swept-source OCT revealed slight hyperreflectivity of the outer nuclear and plexiform layers and disruption of the ellipsoid zone (Figure 4). She was diagnosed with unilateral AMN OS. Treatment was not reported. A pair of unilateral AMN cases were likewise reported in women aged 22 and 28 two days following the AZD1222 vaccine by Mambretti et al. [88]. Chen et al. reported a similar case with similar exam findings in a 21-year-old woman who developed paracentral scotomas three days after receiving her first dose of the BNT162b2 vaccine [89].

A report made by Pichi et al. regarding unilateral AMN OS was unique in that the patient presented after vaccination with the BBIBP-CorV (Sinopharm) vaccine [90]. Furthermore, the patient presented with visual acuity of 20/400—considerably worse than other cases—despite having similar OCT findings as in previous cases. The authors reported that, with observation only, the patient demonstrated significant improvement back to a baseline visual acuity of 20/30.

Michel et al. reported AMN in a 21-year-old woman two days after receiving the AZD1222 vaccine but reported that the patient initially presented with four central scotomas OS, a greater number than in other reports we reviewed. With observation only, however, there was loss of hyperreflectivity of her lesions four days after presentation and her visual field testing showed improvement four weeks later.

In comparison, Book et al. reported a case of a 21-year-old woman with no past medical history who developed bilateral paracentral scotomas 3 days after receiving the AZD1222 vaccine [91]. Near-infrared imaging and OCT revealed similar lesions as the previous case, but bilaterally. She was diagnosed with bilateral AMN. Druke et al. reported a similar case of a 23-year-old female who developed bilateral paracentral scotomas one day after vaccination with the AZD1222 vaccine [92]. Fundus photography revealed a subtle brownish rimmed lesion parafoveal in the right eye and a blurred lesion nasal to the macula. Near-infrared imaging and OCT imaging confirmed a diagnosis of bilateral AMN. Valenzuela et al. described a similar report of bilateral AMN following vaccination with the second dose BNT162b2 vaccine [93]. In contrast to other reports, the authors reported resolution of symptoms after 7 days with observation only.

### 8.2. Paracentral Acute Middle Maculopathy

Paracentral acute middle maculopathy (PAMM) is an entity similar yet distinct to AMN [94,95,96]. Rahimy et al. previously described PAMM as a more superficial variant of AMN with similar manifestations such as acute paracentral scotomas and similar imaging findings of hyporeflective macular lesions [97,98]. However, PAMM predominantly affects middle-aged men, whereas AMN predominantly affects young women [84,85,97,98]. Furthermore, PAMM lesions manifest as thinning and atrophy of the inner nerve fiber layer, whereas AMN lesions manifest at the junction of the outer plexiform and outer nerve fiber layers [97,98].

Limited reports suggest that vaccination against COVID-19 may have an association with PAMM as well. Pichi et al. reported on a patient who received the BBIBP-CorV (Sinopharm) vaccine, and 20 min later developed persistent tachycardia, systolic hypertension, and concurrent development of an inferior scotoma OS [90]. The patient’s fundus examination revealed a suprafoveal dot hemorrhage OS. OCTA showed an area of flow disturbance superior to the fovea OS, and en face swept-source OCT showed a round area of hyperreflectivity superior to the fovea. A diagnosis of PAMM was made. The authors did not comment on follow-up.

Another report by Vinzamuri et al. described a 35-year-old man who received the first dose of the AZD1222 vaccine and developed reduced brightness of vision in both eyes over four weeks [99]. After receiving his second dose, his symptoms progressed further, and he was examined by an ophthalmologist. Although his acuity was 20/20 in both eyes, OCT of his macula revealed hyperreflective lesions involving the nerve fiber layer, ganglion cell layer, and outer plexiform layer. There was focal loss of the external limiting membrane in both eyes. He was diagnosed with bilateral PAMM and AMN simultaneously. With observation only, there was improvement in the patient’s symptoms at a three-week follow-up, with significant reduction in the number and size of hyperreflectivity lesions.

### 8.3. Other Reports

Other reports of retinal phenomena exist but have been reported infrequently. Multiple reports made mention of retinal detachments. Fowler et al. reported on a 33-year-old man who developed blurry vision three days after receiving his first dose of the BNT162b2 vaccine [100]. The patient was found to have a macular serous detachment of the neurosensory retina and OCT revealed a diagnosis of central serous retinopathy. He was successfully treated over three months with spironolactone 50 mg daily. Khochtali et al. reported a case of foveolitis OS in a 24-year-old woman after her first dose of the BNT162b2 vaccine [101]. Imaging showed diffuse retinal vascular leakage, faint foveal hyperfluorescence and late phase hypofluorescence of the foveal lesion, and granular hyperreflective specks in the inner nuclear layer. She demonstrated improvement of her lesions on a six-week PO prednisolone taper and resolution at three months.

### 8.4. Mechanisms

The most commonly reported retinal adverse event following COVID-19 vaccination on our review was AMN. Previous studies have proposed ischemia of the deep capillary plexus (DCP) in the inner nerve fiber layer as the pathologic mechanism underlying AMN [85,86]. Risk factors for AMN include concurrent viral illness, oral contraceptive use, and vasoactive events such as trauma, dehydration, and shock [85]. Of note, oral contraceptive use was described in all cases of AMN we received.

Reports connecting AMN and vaccination are limited. However, Shah et al. and Liu et al. have previously described AMN following influenza vaccination in women aged 42 and 47, respectively [102,103]. In both cases, there was a demonstration of reduced DCP flow on OCTA imaging at baseline and restoration of flow at follow-up.

Mambretti et al. proposed in their reports of AMN that the pro-inflammatory state following COVID-19 vaccination may have had a compounding effect on the pro-thrombotic effects of oral contraceptives [88]. Furthermore, they proposed that hypovolemia associated with the inflammatory reaction following vaccination may have led to reduced blood flow in the DCP. Only the reports by Bøhler et al. and Pichi et al. commented on imaging findings in the retinal vasculature, but both mentioned compromised flow in the DCP [87,90].

While the exact mechanism remains unclear, Giacuzzo et al. also reported development of bilateral AMN in the setting of COVID-19 *infection* with reduced flow in the DCP and unusually large, confluent lesions bilaterally compared to the usual smaller, petaloid lesions [104]. Future investigation should examine whether there is a unifying immunologic explanation underlying COVID-19 vaccination, infection, and development of AMN.

## 9. Vascular

Previous discussion around COVID-19 vaccination has raised the question of whether or not vaccines to COVID-19 confer upon patients a hypercoagulable state [105,106,107]. We reviewed 4 reports (5 patients, 6 eyes) commenting on vascular events following COVID-19 vaccination. Of these 5 patients, 4 (80.0%) were male and 1 (20.0%) was female. The mean age at the time of presentation was 54.6 ± 18.1 years, and the average time from vaccination to development of ophthalmic symptoms was 5.4 ± 6.6 days. One patient (20.0%) presented with bilateral involvement. All 6 eyes had both presenting and final visual acuities reported; the means, respectively, were logMAR 0.676 ± 0.797 (20/95 in Snellen notation) and 0.016 ± 0.036 (20/21) (*p* = 0.128).

### 9.1. Central and Hemi-Retinal Vein Occlusion

Two reports of central retinal vein occlusion (CRVO) have been made following vaccination against COVID-19. Bialasiewicz et al. reported on a 50-year-old male who developed immediate bilateral retrobulbar pain, red eye, and vision loss 15 min after receiving the BNT162B2 [108]. Fundus exam revealed a hemorrhagic CRVO with ischemic areas on fluorescein angiography and OCT showed cystoid macular edema. The patient responded well to three days of acetylsalicylic acid 100 mg daily. Elsewhere, Endo et al. reported a similar case of a 52-year-old male developing sudden blurred vision in the left eye 14 days after his first dose of the BNT162b2 vaccine [109]. His visual acuity at presentation was 20/20 OS. On examination, he had dot hemorrhages in the upper quadrants, dilated tortuous veins in four quadrants, and exudates. Fluorescein angiogram was consistent with non-ischemic CRVO. In this case, the patient demonstrated improvement with intravitreal bevacizumab and PO apixaban.

Tanaka et al. also reported on a pair of cases of unilateral branch retinal vein occlusion (BRVO) exacerbation in a 71-year-old woman (OS) and 72-year-old man (OD), both after the BNT162b2 vaccine [110]. Both patients presented with visual disturbance. Ultra-wide-field pseudo-color and OCT imaging demonstrated recurrence of previously resolved BRVO and macular edema. In the former case, the patient was treated with one-time intravitreal aflibercept, and in the latter, the patient received two doses of intravitreal ranibizumab. Both patients demonstrated resolution of symptoms and macular edema.

One report of hemi-retinal vein occlusion (HRVO) has been made following COVID-19 vaccination. Goyal et al. reported on a 28-year-old man who developed visual disturbances OD following his second dose of the Gam-COVID-Vac/Sputnik V vaccine [111]. Examination and imaging revealed a superior HRVO with severe cystoid macular edema. He demonstrated significant resolution of macular edema within one week on a PO prednisolone taper and apixaban twice daily.

### 9.2. Mechanisms

For a discussion of possible mechanisms underlying hypercoagulable states following vaccination, see Section 5.3.

## 10. Neuro-Ophthalmology

On our review, 4 reports (5 patients, 8 eyes) have discussed neuro-ophthalmological manifestations following COVID-19 vaccination. All 5 patients were female. The mean age at the time of presentation was 48.0 ± 21.5 years, and the average time from vaccination to development of ophthalmic symptoms was 8.6 ± 8.3 days. Three patients (60.0%) presented with bilateral involvement. For the 8 affected eyes which had presenting visual acuities, the mean visual acuity was logMAR 0.732 ± 0.700 at presentation (20/108 in Snellen notation). For the 4 eyes which had both presenting and final visual acuities reported, the mean visual acuities were 0.490 ± 0.412 and 0.024 ± 0.042, respectively (20/61 and 20/21 in Snellen notation; *p* = 0.166).

### 10.1. Optic Neuritis

The majority of the reports that we reviewed dealt with cases on the axis of optic neuritis. Elnahry et al. reported on a pair of cases [112]. A 69-year-old woman presented with blurry vision in both eyes with examination revealing optic nerve head edema bilaterally. OCT showed swelling of the retinal nerve fiber layer in both eyes with intraretinal and subretinal fluid in the right macula. The patient was diagnosed with post-vaccination central nervous system (CNS) inflammatory syndrome leading to neuroretinitis and papillitis. They also reported on a 32-year-old female who similarly presented with left optic neuritis. Both cases demonstrated significant improvement in symptoms and examination with use of IV methylprednisolone.

Pawar et al. reported on a 28-year-old female who developed sudden vision loss OS three weeks after receiving an unspecified COVID-19 vaccination [113]. Examination and imaging were consistent with left optic neuritis. As in the previous cases, the patient developed resolution of symptoms after IV methylprednisolone followed by PO steroids.

Leber et al. reported on a 32-year-old woman who developed rapidly progressive worsening vision and pain with extraocular movements OS [114]. Presenting visual acuity was 20/20 OD and 20/200 OS. Examination revealed and RAPD OS as well as disc swelling OD and OS. MRI revealed bilateral optic neuritis and labs also revealed a thyroid-stimulating hormone (TSH) level of 13.2 mUI/L (reference 0.45–4.5 mUI/L). The patient demonstrated significant improvement in symptoms and examination following five days of IV methylprednisolone 1 g.

### 10.2. Other Reports

Maleki et al. reported on a neuro-ophthalmologic case as well. A 79-year-old woman presented with bilateral sudden loss of vision 2 days after receiving the BNT162b2 vaccine [53]. Her examination was significant for an APD OD with generalized disc pallor OD and inferior pallor OS (Figure 5). A diagnosis of arteritic anterior ischemic optic neuropathy (AAION) was made.

### 10.3. Mechanisms

We have previously discussed the mechanisms underlying optic neuropathy in the setting of vaccination. While the phenomenon is poorly studied, it has been previously posited that molecular mimicry between myelin basic protein and viral particles, epitope spreading, bystander activation, and superantigen activation all may play a role in the development of autoimmune optic neuritis following vaccination [14,115,116,117,118,119].

## 11. Ocular Motility Disorders

Multiple reports of gaze palsies have been made. On our review, 7 reports (9 patients, 10 eyes) discussed gaze palsies. One case was bilateral. Of these cases, six (66.67%) were male and three (33.3%) were female. The mean age at the time of presentation was 45.3 ± 18.7 years, and the mean time between vaccination and development of ophthalmic symptoms was 8.8 ± 9.4 days. Visual acuities were discussed sparingly.

### 11.1. Ocular Gaze Palsies

Pawar et al. reported on a series of three cases of gaze palsies [113]. They reported on a 24-year-old woman who developed diplopia 21 days after unspecified COVID-19 vaccination. On examination, she demonstrated restricted elevation of both eyes. MRI and neurological examination were otherwise normal. The patient was diagnosed with bilateral vertical gaze palsy. The patient underwent systemic steroid therapy and demonstrated resolution after 10 days.

Pawar et al. also reported on a case of acute abducens (6th cranial) nerve palsy in the left eye of a 44-year-old male who developed his symptoms 28 days after unspecified vaccination against COVID-19. The patient had normal examination and imaging findings otherwise. The patient was treated with Botox injection into the medial rectus OS and thereafter had only minimal residual esotropia. Finally, they reported on a patient who developed acute onset esotropia OS 6 days after receiving vaccination against COVID-19. The patient previously had recurrent abducens nerve palsy following a chickenpox infection. The authors did not report on treatment. Elsewhere, Reyes-Capo et al. also reported a similar case of abducens nerve palsy in a 59-year-old woman who presented with new esotropia and abduction deficits OD after her first dose of the BNT162b2 vaccine [120]. Examination remained unchanged on follow-up. Kawtharani et al. also reported on a case of abducens nerve palsy OS after the AZD1222 vaccine which resolved with vitamin B12 supplementation and eyeglasses modification, but the patient also went on to develop transverse myelitis and quadriplegia in the ensuing weeks [121].

In another report, Pappaterra et al. reported on an 81-year-old male who developed acute diplopia one day after receiving the first dose of the Moderna COVID-19 vaccine [122]. Examination revealed limited adduction and infraduction OS only with normal motility OD. Further examination revealed an exotropia of 3 prism diopters (PD) in the primary position, decreasing to an exotropia of 2 PD on right gaze and increasing to an exotropia of 10 PD on left gaze, combined with a left hypertropia of 5 PD. Ultimately, the patient was diagnosed with a partial left oculomotor (3rd cranial) nerve palsy. At an 11 day follow up, the patient had full extraocular motility in both eyes with minimal residual exodeviation in the primary position with observation only.

Manea et al. reported on a 29-year-old man who developed multiple cranial neuropathies six days after receiving his first dose of the BNT162b2 vaccine [123]. In particular, he developed incomplete oculomotor (3rd cranial), abducens (6th cranial), and facial (7th cranial) nerve palsies.

Eleiwa et al. reported on a 46-year-old man who developed torsional, binocular diplopia three days after receiving the second dose of the AZD1222 vaccine [14]. Examination was consistent with a diagnosis of right trochlear (4th cranial) nerve palsy.

### 11.2. Ocular Gaze Palsies

Vaccination-induced cranial nerve palsies have previously been reported following the influenza, hepatitis B, smallpox, and MMR vaccines [124,125,126,127,128,129,130]. While the exact mechanism has not been elucidated, vaccine-induced cranial nerve palsies are believed to be due to immune-mediated damage resulting in demyelination or vascular compromise resulting in reduced blood flow [130,131]. Given the aforementioned links between vaccination and vascular compromise as well as autoimmune phenomena, it is likely that these underlie the development of ocular motility disorders following COVID-19 vaccination.

## 12. Other Reports

Only one report of scleritis was found on our review. Pichi et al. reported on a patient with unspecified age, gender, and vaccine who presented seven days after vaccination with bilateral eye redness and pain [90]. The patient’s ocular exam was significant for scleral hyperemia and positive phenylephrine test results. No AC cells or flare was present. A diagnosis of scleritis was made. The patient was thereafter successfully treated over one week with a topical steroid taper which resulted in complete resolution.

Jumroendararasame et al. reported on a 42-year-old man who developed blurred vision ten minutes after receiving the Corona Vac [132]. The patient, who was himself an ophthalmologist, described immediate blurred vision centrally which was followed thereafter by obscuration of the visual field OS. Examination and OCT imaging were unrevealing. The patient experienced resolution of his symptoms approximately two hours after the initial event. The authors proposed that acute vasospasm in the vasculature of the postchiasmatic visual pathway was the underlying cause of the event.

Santovito and Pinna previously reported on a male patient who developed sudden darkening of his visual field after receiving the BNT162b2 vaccine but were unable to comment on an etiology [133]. We have previously discussed this case elsewhere, suggesting that it may have been an optic neuropathy [14].

## 13. Conclusions

Since the introduction of COVID-19 vaccinations, numerous reports have commented on adverse ocular events following vaccination. In this review, we sought to present these in a systematic fashion and offer insights into the mechanisms and clinical considerations surrounding these phenomena. Given the relatively low number of reports per specific phenomenon, more reports and clinical data are needed in order to establish better guidelines and insights. Leading clinical bodies in ophthalmology have not provided official guidelines on the use of COVID-19 boosters in the setting of active ophthalmic complications, yet it is our belief that caution and delay may be warranted in lower-risk groups with active complications. Nevertheless, it is the opinion of the authors that vaccination is a vital public health tool in the management of the COVID-19 pandemic.

## Figures and Tables

**Figure 1 vaccines-10-00342-f001:**
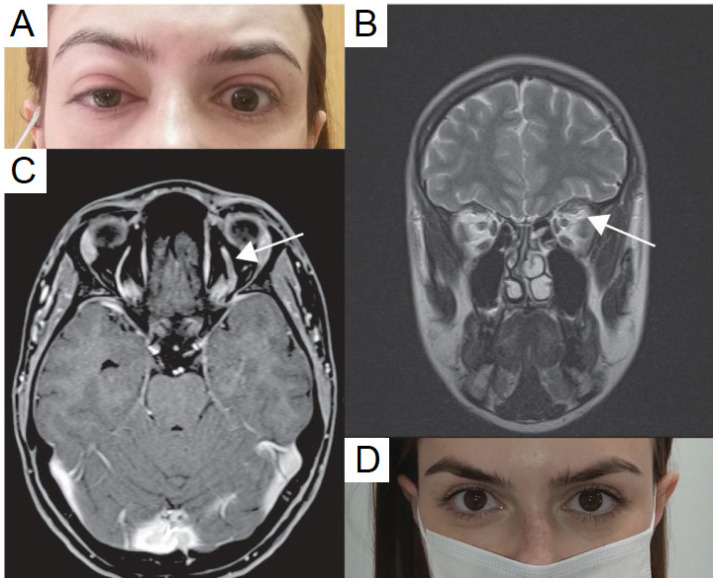
Clinical presentation of the vaccine-induced prothrombotic immune thrombocytopenic disorder (VIPIT) and superior ophthalmic vein (SOV) thrombosis after ChAdOx1 nCoV-19 vaccination. (**A**) patient presentation at admission with marked proptosis, (**B**) contrast enhanced magnetic resonance imaging (MRI) revealed SOV thrombosis (white arrow), presented with widening SOV and filling defects, (**C**) T2 sequence further confirmed SOV thrombosis with the enhanced signal intensity of SOV (white arrow), (**D**) no symptoms after five days of treatment, published with patient’s permission. Adapted from Panovska-Stavridis, I.; Pivkova-Veljanovska, A.; Trajkova, S.; Lazarevska, M.; Grozdanova, A.; Filipche, V. A Rare Case of Superior Ophthalmic Vein Thrombosis and Thrombocytopenia Following ChAdOx1 nCoV-19 Vaccine Against SARS-CoV-2. *Mediterr. J. Hematol. Infect. Dis.*
**2021**, *13*, e2021048; Published 1 March 2021. https://doi.org/10.4084/MJHID.2021.048 [20]. Figure 1, Copyright (2021) with permission from Institute of Hematology, Catholic University, Rome, open access article under the terms of the Creative Commons Attribution License.

**Figure 2 vaccines-10-00342-f002:**
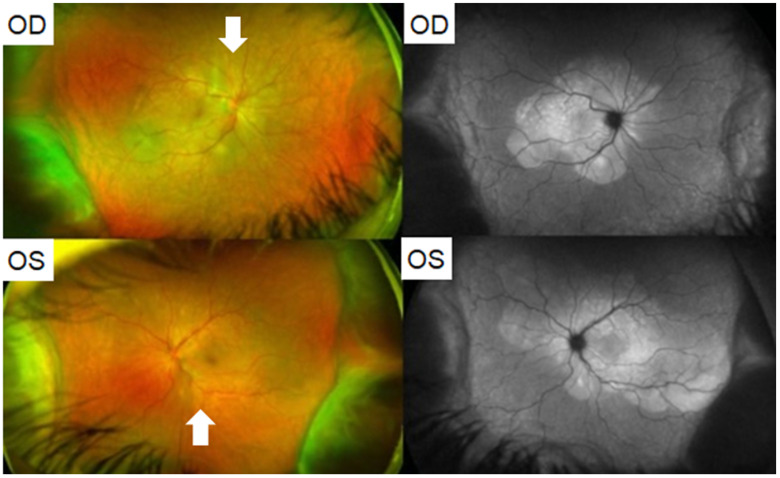
(Patient 1) Fundus photography and autofluorescence of both eyes showing serous retinal detachment (white arrows) and optic disc hyperemia. Adapted from Saraceno, J.J.F.; Souza, G.M.; Dos Santos Finamor, L.P.; Nascimento, H.M.; Belfort, R., Jr.; Vogt-Koyanagi-Harada Syndrome following COVID-19 and ChAdOx1 nCoV-19 (AZD1222) vaccine. *Int. J. Retina Vitreous*. **2021**, *7*, 49; Published 30 August 2021. https://doi.org/10.1186/s40942-021-00319-3 [51]. Figure 1 Copyright (2021) with permission from Springer Nature, open access article under the terms of the Creative Commons Attribution License.

**Figure 3 vaccines-10-00342-f003:**
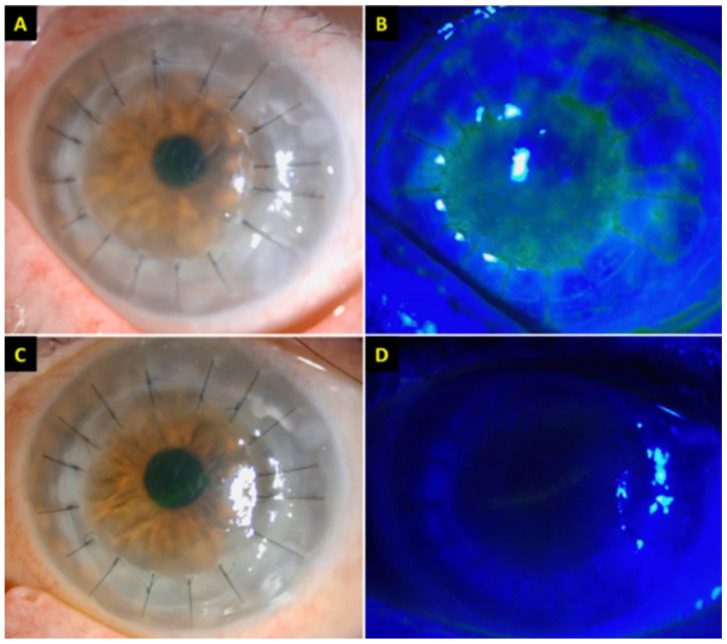
(**A**,**B**) Slit-lamp photography demonstrating conjunctival hyperemia, corneal graft haze, diffuse corneal epithelial, and stromal oedema (within the graft), Descemet’s folds, scattered keratic precipitates (KPs), and 1+ cells in anterior chamber. An unusual distribution of fluorescein staining with coarse punctate epitheliopathy over the corneal graft was observed. The central corneal thickness (CCT) was 730 μm. (**C**,**D**) At 3-week post treatment, the corneal graft rejection was successfully treated with considerable improvement in the graft transparency, reduction in epithelial and stromal oedema, and resolution of epitheliopathy and anterior chamber inflammation. The best-corrected visual acuity improved to 6/12, with a CCT of 609 μm. Adapted from Rallis, K.I.; Ting, D.S.J.; Said, D.G.; et al. Corneal graft rejection following COVID-19 vaccine. Eye (2021). https://doi.org/10.1038/s41433-021-01671-2 [65]. Figure 1, Copyright (2021) with permission from Nature Publications, open access article under the terms of the Creative Commons Attribution License.

**Figure 4 vaccines-10-00342-f004:**
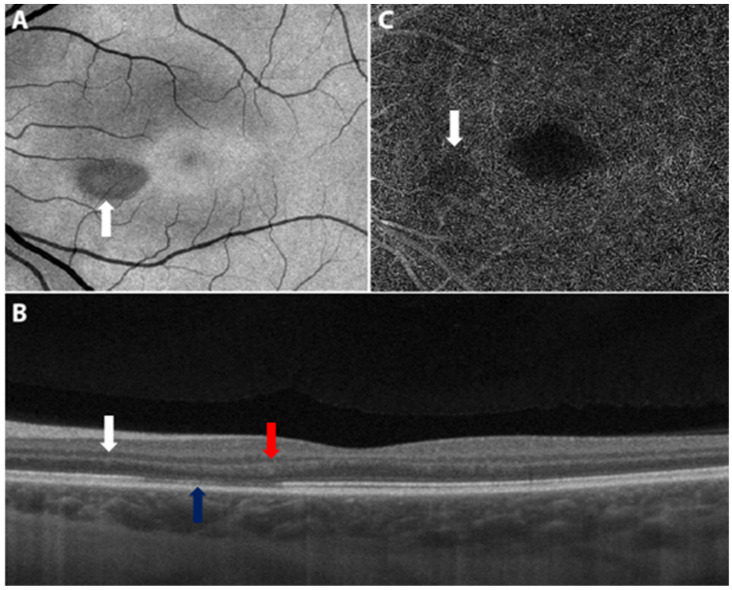
Swept source optical coherence tomography of the left macula. (**A**) The en face image displays a teardrop-shaped macular lesion (white arrow) nasally to the fovea. (**B**) The cross-sectional image displays slight hyperreflectivity of the outer nuclear (white arrow) and plexiform (red arrow) layers and disruption of the ellipsoid zone (blue arrow) corresponding to the lesion. (**C**) The angiogram indicates subtle dropout (white arrow) in the deep capillary plexus corresponding to the lesion. Adapted from Bøhler, A.D.; Strøm, M.E.; Sandvig, K.U.; et al. Acute macular neuroretinopathy following COVID-19 vaccination. Eye (2021). https://doi.org/10.1038/s41433-021-01610-1 [87]. Figure 2, Copyright (2021) with permission from Springer Nature, open access article under the terms of the Creative Commons Attribution License.

**Figure 5 vaccines-10-00342-f005:**
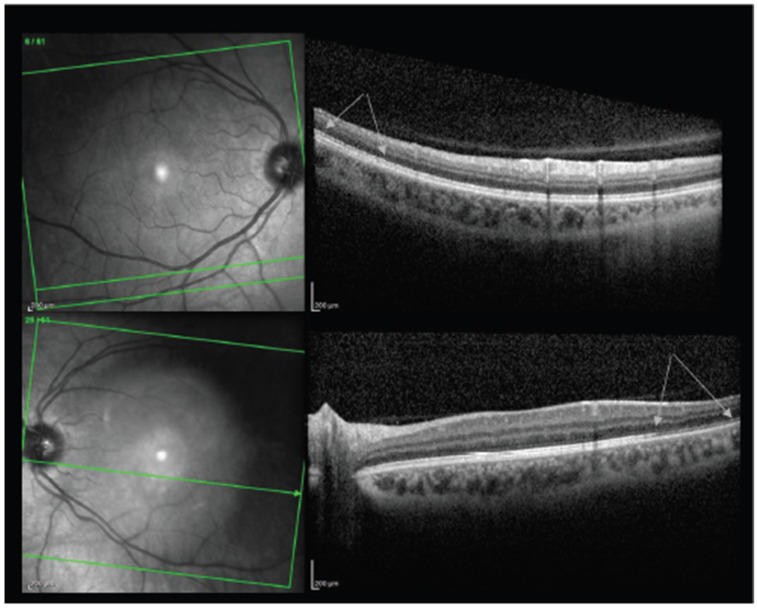
The upper and lower pictures are macular optical coherence tomography of the right and left eye, respectively. Arrows show the areas of disruption and segmentation of the ellipsoid zone in the right eye and thinning of (absent in some areas) ellipsoid zone in the left eye. Adapted from Maleki A, Look-Why S, Manhapra A, Foster CS. COVID-19 Recombinant mRNA Vaccines and Serious Ocular Inflammatory Side Effects: Real or Coincidence? *J. Ophthalmic. Vis. Res*. **2021**, *16*, 490–501; Published 29 July 2021. https://doi.org/10.18502/jovr.v16i3.9443 [53]. Figure 4, Copyright (2021) with permission from KnE Publishing, open access article under the terms of the Creative Commons Attribution License.

**Table 1 vaccines-10-00342-t001:** Aggregated information on reviewed cases.

Author	Age	Sex	Vaccine	Time from Vaccine to Symptom (days)	Presenting VA	Side	Manifestations
Eyelid							
Austria et al., 2021	32	F	BNT162b2, #NR	1 to 2	NR	NR	Unilateral upper greater than lower eyelid edema and erythema without other systemic or ocular findings on exam.
	43	F	BNT162b2, #NR				
	43	F	BNT162b2, #NR				
Mazzatena et al., 2021	67	F	BNT162b2, #1	10	NR	OD and OS	Ecchymotic lesions on the upper eyelids. Lesions were moderately itchy.
	44	F	BNT162b2, #2	21	NR	OD and OS	Purpuric lesions bilaterally. Lesions were circumscribed on the upper eyelid and totally asymptomatic.
	63	M	BNT162b2, #2	21	NR	OD and OS	Purpuric lesions bilaterally. Lesions were circumscribed on the upper eyelid and totally asymptomatic.
Orbit							
Bayas et al., 2021	55	F	AZD1222, #1	10	20/140	OD	Bilateral conjunctival congestion, retroorbital pain, and diplopia. MRI showed bilateral superior ophthalmic vein thrombosis.
					20/140	OS	
Chuang et al., 2021	45	M	NR	7	NR	OS	Progressive ptosis and decreased vision OS, diplopia, and examination with APD and complete ophthalmoplegia. CT and MRI with left cavernous sinus thrombosis. Pt diagnosed with Tolosa-Hunt syndrome.
Panovska-Stavridis et al., 2021	29	F	AZD1222, #1	10	NR	OS	Left orbital swelling, severe headache, and blurred vision OS. Labs showed thrombocytopenia of 18 × 10^19^/L. MRI demonstrated central filling defects and a diagnosis of superior ophthalmic vein thrombosis was made.
Cornea							
De la Presa et al., 2021	27	F	Moderna Vaccine, #1	15	20/20	OD	Redness and irritation with 1+ conjunctival hyperemia and an irregular temporal epithelial rejection line in a patient post LR-CLAL 4 years earlier. A diagnosis of acute unilateral graft rejection was made.
Abousy et al., 2021	73	F	BNT162b2, #2	4	20/200	OD	Vision loss with corneal thickening with Descemet folds bilaterally in a patient with DSEK 8 years previously, consistent with acute bilateral graft rejection.
					20/40	OS	
Crnej et al., 2021	71	M	BNT162b2, #1	7	20/125	OD	Painless decrease in right eye vision with conjunctival injection and diffuse corneal edema 5 months post-DMEK, diagnosed as acute unilateral graft rejection.
Khan et al., 2021	48	M	AZD1222, #1	21	LP	OD	Vision loss, bilateral lid edema, diffuse conjunctival and ciliary congestion, corneal melting and perforation with diffuse corneal haze, uveal tissue prolapse, bilateral massive choroidal detachment on B-scan ultrasonography.
					LP	OS	
Nioi et al., 2021	44	F	BNT162b2, #1	13	CF	OS	Blurry vision, eye redness and discomfort OS. Examination with ciliary injection, diffuse corneal edema, keratic precipitates, Descemet folds, anterior chamber cells, consistent with acute unilateral graft rejection.
Papasavvas et al., 2021	69	F	BNT162b2, #1	10	20/30	OD	Excruciating pain in the left V1 dermatome with a small dendrite in the supero-temporal cornea. Diagnosis of HZO was made.
	73	F	BNT162b2, #3	16	20/40	OD	Excruciating pain in the right V1 dermatome without dendrite formation. Vitreous cells present. Diagnosis of HZO was made.
	72	F	Moderna Vaccine, #1	13	20/63	OS	Excruciating pain in the left V1 dermatome with conjunctival chemosis but no corneal or AC changes. 10 days later with AC uveitis with cell, flare, KP, and Descemet folds. Diagnosis of HZO was made.
Parmar et al., 2021	35	M	AZD1222, #1	2	CF	OS	Decreased vision in a patient post-repeat PKP 6 months previously after original PKP 3 years earlier. Exam with graft edema more prominent in the lower half as well as KPs and AC reaction. Diagnosis of acute unilateral graft rejection was made.
Phylactou et al. 2021	66	F	BNT162b2, #1	7	20/125	OD	Acute-onset right eye blurred vision, redness, and photophobia with conjunctival injection, diffuse corneal edema, fine KP, 1+ AC cells 21 days post-DMEK, diagnosed as acute unilateral graft rejection.
	83	F	BNT162b2, #2	21	20/80	OD	Acute-onset bilateral blurred vision, pain, photophobia and red with bilateral circumcorneal injection, KP, and AC inflammation, 6 (OD) and 3 (OS) years post-DMEK, diagnosed as acute bilateral graft rejection.
					20/40	OS	
Rallis et al., 2021	68	F	BNT162b2, #1	3	CF	OS	Vision loss OS with conjunctival hyperemia, diffuse corneal punctate staining and graft edema, and KP 3 months post-redo PKP for failed DSAEK, diagnosed as acute unilateral graft rejection. Pre-existing OD graft was intact.
Ravichandran and Natarajan 2021	62	M	AZD1222, #1	21	NR	NR	Right eye decreased vision and congestion, with an advancing Kodadoust rejection line and corneal graft edema, 2 years post-PKP. Diagnosed as acute unilateral graft rejection.
Wasser et al., 2021	73	M	BNT162b2, #1	13	20/200	OS	Eye discomfort OS and vision loss with ciliary injection, corneal edema, Descemet folds, and KP 2 years after re-graft for PKP performed 44 years earlier. Diagnosed as acute unilateral graft rejection.
	56	M	BNT162b2, #1	12	CF	OD	Blurred vision and redness OD, with diffuse corneal edema, KP, and AC cells 25 years post-PKP, diagnosed as acute unilateral graft rejection. Pre-existing OS graft from PKP 7 years earlier was intact.
Uveitis							
ElSheikh et al., 2021	18	F	Sinopharm, #2	5	20/40	OD	Bilateral acute uveitis with 2+ AC flare OU and 1+ cell OU and hyperreflective dots in the AC in a patient with juvenile idiopathic arthritis.
					20/120	OS	
Goyal et al., 2021	34	M	AZD1222, #1	4	20/120	OD	Ocular pain followed by nasal redness OS and a floater OD progressing to severe vision loss. Fundus exam with multiple bilateral oval lesions at the level of the choroid with serous detachments, consistent with bilateral multifocal choroiditis.
					20/20	OS	
Herbort and Papasavvas 2021	53	M	Moderna Vaccine, #2	5	NR	OD	Severe flare-up of pre-existing herpes-keratouveitis OD inactive for 18 months without treatment. Pt presented with numerous KPs, elevated IOP to 41 mmHg.
Ishay et al., 2021	28	M	BNT162b2, #1	10	NR	OS	Pain, redness, and blurred vision OS in a patient with Behçet’s disease on colchicine twice daily. Examination revealed severe panuveitis.
Jain and Kalamkar 2021	27	M	AZD1222, #1	2	20/20	OS	Pain, redness and severe circumcorneal congestion OS with 2+ AC cells and non-granulomatous KP in a patient with juvenile idiopathic arthritis and one previous episode of bilateral uveitis. Acute uveitis was diagnosed.
Koong et al., 2021	54	M	BNT162b2, #1	1	20/80	OD	Acute bilateral, sequential blurring of vision with bilateral areas of subretinal fluid with dot-blot hemorrhages on examination. OCT with bilateral serous neurosensory retinal detachments. ICGA confirmed diagnosis of VKH.
					20/160	OS	
Maleki et al., 2021	33	F	Moderna Vaccine, #2	10	20/20	OD	Bilateral photopsia and progressive nasal field defect OS. OCT with outer layer segmental disruption OS. Elevated ESR and CRP. Diagnosis of acute zonal occult outer retinopathy (AZOOR) was made.
					20/20	OS	
Mishra et al., 2021	71	M	AZD1222, #1	10	CF	OD	Reactivation of VZV presenting with panuveitis OD, circumcorneal congestion, multiple fine keratic precipitates, anterior chamber cells and flare, vitritis, and widespread acute retinal necrosis.
Mudie et al., 2021	43	F	BNT162b2, #2	3	20/500	OD	Bilateral substantial vision loss, eye pain and redness, and photophobia, with 3–4+ AC cell and 2–3+ vitreous cell. OCT with significant choroidal thicnening, FA with mild peripheral vascular leakage. Diagnosis of panuveitis was made.
					20/500	OS	
Pan et al., 2021	50	F	Unspecified inactivated Vero cell-based vaccine approved in China	5	20/33	OD	Bilateral blurred vision with pale, blurry optic disc, absent foveal reflex, macular edema, and fluorescein angiography consistent with bilateral choroiditis.
					20/66	OS	
Papasavvas and Herbort 2021	43	F	BNT162b2, #2	42	20/20	OD	Reactivation of pre-existing VKH disease with significant anterior segment inflammation OU, and 3–4 mutton-fat KP OD. OCT showed retinal folds and subretinal fluid. Multiple hypofluorescent dark dots present on ICGA.
					20/20	OS	
Rabinovitch et al., 2021	43	F	BNT162b2, #1	2	20/25	OD	Redness, pain, blurred vision. 3+ cell and 1+ flare and fibrin on exam. Diagnosis of anterior uveitis was made.
	34	M	BNT162b2, #1	4	20/32	OD	Redness and pain. 1+ cell and non-granulomatous KPs on exam. Diagnosis of anterior uveitis was made.
	34	F	BNT162b2, #1	1	20/50	OS	Redness, pain, and photophobia. 2+ cell and non-granulomatous KPs on exam. Diagnosis of anterior uveitis was made.
	53	M	BNT162b2, #1	13	20/25	OS	Pain only. 0.5+ cell on exam. Diagnosis of anterior uveitis was made.
	64	M	BNT162b2, #1	15	20/25	OS	Redness, pain, and photophobia. 0.5+ cell on exam. Diagnosis of anterior uveitis was made.
	68	M	BNT162b2, #1	5	20/200	OD	Redness and pain. 1+ cell on exam. Diagnosis of anterior uveitis was made.
	61	F	BNT162b2, #1	12	20/25	OD	Pain and photophobia. 2+ cell on exam. Diagnosis of anterior uveitis was made.
	65	F	BNT162b2, #1	3	20/80	OD	Redness, pain, photophobia, and blurred vision. 2+ cell and 2+ flare on exam. Diagnosis of anterior uveitis was made.
	78	M	BNT162b2, #2	3	20/25	OS	Redness, pain, and blurred vision. 2+ cell and 2+ flare with posterior synechiae on exam. Diagnosis of anterior uveitis was made.
	59	M	BNT162b2, #2	8	20/32	OS	Pain, photophobia, and blurred vision. 2+ cell on exam. Diagnosis of anterior uveitis was made.
	72	M	BNT162b2, #2	16	20/80	OD	Redness only. 1+ cell on exam. Diagnosis of anterior uveitis was made.
	51	M	BNT162b2, #2	2	20/50	OS	Redness and pain. 2+ cell on exam. Diagnosis of anterior uveitis was made.
	42	F	BNT162b2, #2	20	20/25	OD	Pain and blurred vision bilaterally. 2+ cell on exam. Diagnosis of anterior uveitis was made.
					20/25	OS	
	74	M	BNT162b2, #2	7	20/40 OD	OD	Pain only. 1+ cell and 2+ flare on exam. Diagnosis of anterior uveitis was made.
	39	M	BNT162b2, #2	5	20/32	OD	Blurred vision with defect and photopsia. Outer retinal changes on exam. Diagnosis of MEWDS was made.
	64	F	BNT162b2, #2	6	20/25	OD	Photophobia only. 1+ flare on exam. Diagnosis of anterior uveitis was made.
	50	F	BNT162b2, #2	2	20/25	OS	Pain only. 1+ cell on exam. Diagnosis of anterior uveitis was made.
	23	F	BNT162b2, #2	2	20/25	OD	Redness, blurred vision, and photophobia bilaterally. 1+ cell and 1+ flare on exam. Diagnosis of anterior uveitis was made.
					20/25	OS	
	36	M	BNT162b2, #2	1	20/80	OS	Redness, photophobia, and blurred vision. 3+ cell and 3+ flare with non-granulomatous KPs on exam. Diagnosis of anterior uveitis was made.
	41	M	BNT162b2, #2	2	20/50	OD	Redness, photophobia, and blurred vision. 2+ cell and 2+ flare on exam. Diagnosis of anterior uveitis was made.
	28	F	BNT162b2, #2	30	20/32	OS	Blurred vision, visual field defect, and photopsia. Outer retinal changes on exam. Diagnosis of MEWDS was made.
Renisi et al., 2021	23	M	BNT162b2, #2	14	20/40	OS	Pain and photophobia OS with perikeratic and conjunctival hyperemia, posterior synechiae, AC cells, and KP. Diagnosis of anterior uveitis was made.
Saraceno et al., 2021	62	F	AZD1222, #1	2	20/600	OD	Acute bilateral loss of vision with mild 2+ AC cell and 1+ vitreous cell OU. Fundus examination revealed a serous retinal detachment OU. OCT revealed the same and subretinal hyperreflective dots. Diagnosis of VKH was made.
					20/200	OS	
Retina							
Bøhler et al., 2021	27	F	AZD1222, #1	2	20/20	OS	Left eye paracentral scotoma with a teardrop-shaped macular lesion nasal to the fovea on ophthalmoscopy, diagnosed as unilateral AMN.
Book et al., 2021	21	F	AZD1222, #1	3	20/16	OD	Bilateral paracentral scotomas with underlying circumscribed paracentral dark lesions on exam, OCT with outer plexiform layer thickening and discontinuity, diagnosed as bilateral AMN.
					20/16	OS	
Chen et al., 2021	21	F	BNT162b2, #1	3	20/20	OS	Paracentral scotomas OS with barely visible oval parafoveal lesions on fundus exam. Infrared imaging revealed hypo-reflective lesions consistent with left AMN.
Drüke et al., 2021	23	F	AZD1222, #1	1	20/20	OD	Development of bilateral paracentral scotomas. Fundus photography revealed a subtle brownish rimmed lesion parafoveally OD and blurred lesion nasal to the macula OS. IR and OCT imaging confirmed a diagnosis of AMN.
					20/20	OS	
Fowler et al., 2021	33	M	BNT162b2, #1	3	20/63	OD	Blurry vision OD with swollen macula, central foveal thickness (CFT) of 457 μm on OCT, and macular serous detachment of the neurosensory retina on FA. OCTA confirmed a diagnosis of central serous retinopathy.
Khochtali et al., 2021	24	F	BNT162b2, #1	5	20/40	OS	Foveolitis with 2+ vitreous cell, diffuse retinal vascular leakage, faint foveal hyperfluorescence and late phase hypofluorescence of the foveal lesion, and granular hyperreflective specks in the inner nuclear layer.
Mambretti et al., 2021	22	F	AZD1222, #1	2	20/20	OD	Acute paracentral scotoma OD with barely visible parafoveal lesions on fundus exam. OCT was consistent with AMN.
	28	F	AZD1222, #1	2	20/20	OD	Acute paracentral scotoma OD with OCT consistent with AMN.
Michel et al., 2021	21	F	AZD1222, #1	2	20/20	OS	Acute-onset of 4 central scotomas OS, well-demarcated dark oval-shaped areas surrounding the left fovea on infrared imaging. OCT with multifocal highly reflective lesions and with ellipsoid and interdigitation zone disruption consistent with AMN.
Pichi et al., 2021	NR	NR	Sinopharm, #NR	5	20/400	OS	Acute vision loss OS, with OCT showing hyperreflectivity of the outer plexiform, Henle fiber, and outer nuclear layers. A diagnosis of AMN was made.
	NR	NR	Sinopharm, #NR	0	20/30	OS	Tachycardia, systolic hypertension (210 mm Hg), and inferior scotoma OS 20 min after vaccination. Fundus examination revealed a suprafoveal dot hemorrhage. A diagnosis of PAMM was made.
Subramony et al., 2021	22	F	Moderna Vaccine, #2	10	20/70	OD	Progressive painless vision loss OD and no vision changes OS, but macula-off inferotemporal retinal detachment OD and small macula-on temporal retinal detachment OS.
					20/20	OS	
Valenzuela et al., 2021	20	F	BNT162b2, #2	2	20/20	OD	Development of bilateral paracentral scotomas and shimmering lights. Fundus exam was unrevealing, but OCT demonstrated corresponding parafoveal foci of hyperreflectivity. Diagnosis of AMN was made.
					20/20	OS	
Vinzamuri et al., 2021	35	M	AZD1222, #2	NR	20/20	OD	Visual disturbance, OCT with hyperreflective lesions involving the nerve fiber layer, ganglion cell layer and outer plexiform layer; diagnosed as PAMM and AMN.
					20/20	OS	
Vascular							
Bialasiewicz et al., 2021	50	M	BNT162b2, #2	0	CF	OD	Immediate bilateral retrobulbar pain, red eye, and vision loss. Examination and OCT revealed a hemorrhagic CRVO with ischemic areas and cystoid macular edema.
					CF	OS	
Endo et al., 2021	52	M	BNT162b2, #1	14	20/20	OS	Sudden blurred vision OS with minimal dot hemorrhages in the upper quadrants, dilated tortuous veins in four quadrants, and disperse exudates. FA was consistent with non-ischemic CRVO.
Goyal et al., 2021	28	M	Sputnik V, #2	11	20/30	OD	Visual disturbance with fundus examination revealing superior hemi-retinal vein occlusion with severe cystoid macular edema.
Tanaka et al., 2021	71	F	BNT162b2, #2	1	20/30	OS	Vision loss, with examination and OCT showing superior temporal BRVO and secondary macular edema with previously resolved inferior temporal BRVO.
	72	M	BNT162b2, #1	1	20/25	OD	Vision loss, with examination and OCT showing recurrence of previously resolved superior temporal BRVO and macular edema.
Neuro-Ophthalmology							
Elnahry et al., 2021	69	F	BNT162b2, #2	16	CF	OD	Blurry vision OU with immediate OS clearing but persistent blurring OD. Examination with optic nerve head edema (OD > OS) and RAPD OD on exam. RNFL imaging confirmed a diagnosis of central nervous system inflammatory syndrome with neuroretinitis.
					20/20	OS	
	32	F	AZD1222, #1	4	20/30	OS	Blurred vision with superior field defect OS. Examination revealed left optic disc swelling and RAPD with decreased RNFL thickness. MRI was diagnostic of left optic neuritis.
Leber et al., 2021	32	F	Corona Vac, #2	0	20/200	OS	Rapidly progressive worsening vision and pain with EOM OS. Examination revealed RAPD OS and disc swelling OD and OS. Labs revealed thyroiditis and MRI revealed bilateral optic neuritis.
					20/20	OD	
Maleki et al., 2021	79	F	BNT162b2, #2	2	20/1250	OD	Bilateral sudden loss of vision, OD > OS, with 3+ afferent pupillary defect OD. OCT, FA, and ICG consistent with generalized disc pallor OD and inferior pallor OS, consistent with bilateral arteritic anterior ischemic optic neuropathy (AAION).
					20/40	OS	
Pawar et al., 2021	28	F	NR	21	20/120	OS	Sudden vision loss OS, with examination revealing mild blurring of the optic disc margin. MRI was consistent with optic neuritis.
Ocular Motility							
Eleiwa et al., 2021	46	M	AZD1222, #2	3	NR	OD	Torsional, binocular diplopia. A diagnosis of right trochlear (4th cranial) nerve palsy was made.
Kawtharani et al., 2021	37	F	AZD1222, #1	NR	NR	OS	Left eye esotropia diagnosed as abducens (6th cranial) nerve palsy.
Manea et al., 2021	29	M	BNT162b2, #1	6	NR	OS	Multiple cranial neuropathies, namely incomplete oculomotor (3rd cranial), abducens (6th cranial), and facial (7th cranial) nerve palsy.
Pawar et al., 2021	23	M	NR	6	NR	OS	Acute esotropia OS in a patient with previous recurrent abducens (6th cranial) nerve palsy following chickenpox. Normal fundus examination and MRI.
	24	F	NR	21	NR	OD	Diplopia and squinting bilaterally, with examination revealing restricted elevation of both eyes. MRI and neurological examination were otherwise normal. Pt was diagnosed with bilateral vertical gaze palsy.
					NR	OS	
	44	M	NR	28	NR	OS	Acute abducens (6th cranial) nerve palsy OS. Normal fundus examination and MRI otherwise.
Pappaterra et al., 2021	81	M	Moderna Vaccine, #1	1	20/30	OS	Acute bilateral oblique diplopia. Examination revealed limited adduction and infraduction OS only. Diagnosis of oculomotor (3rd cranial) nerve palsy was made.
Pereira et al., 2021	65	M	AZD1222, #NR	3	20/20	OD	Sudden-onset painless binocular diplopia, with examination revealing esotropia OD of 12 PD and severe abduction deficit. Diagnosis of right abducens (6th cranial) nerve palsy was made.
Reyes-Capo et al., 2021	59	F	BNT162b2, #1	2	20/25	OD	Acute binocular diplopia and painless, horizontal diplopia, and new right esotropia and abduction deficits OD only. Pt was diagnosed with abducens (6th cranial) nerve palsy.
Other							
Pichi et al., 2021	NR	NR	NR	7	20/20	OD	Bilateral eye redness and pain, with examination demonstrating significant scleral hyperemia with positive phenylephrine test results. No AC cell or flare was present. A diagnosis of scleritis was made.
					20/20	OS	
Santovito and Pinna 2021	NR	M	BNT162b2, #2	NR	NR	OD	Sudden darkening of visual field and reduction of visual acuity preceded hours earlier by unilateral headache and succeeded by confusion and nausea.
					NR	OS	
Jumroendararasame et al., 2021	42	M	Corona Vac, #2	0	20/20	OD	Immediate blurred vision centrally followed by obscuring of the left visual field. Examination and OCT imaging were unremarkable. Authors proposed acute vasospasm as the underlying cause.
					20/20	OS	

CR = case report, LTE = letter to the editor, CS = case series, PE = photo essay, COR = cornea, NEUR = neuro-ophthalmology, ORB = orbital, RET = retina, UVE = uveitis, VASC = vascular, OD = right eye, OS = left eye, OU = both eyes, LR-CLAL = living-relative conjunctival limbal autograft, DMEK = Descemet’s membrane endothelial keratoplasty, HZO = herpes zoster ophthalmicus, AC = anterior chamber, KP = keratic precipitates, PKP = penetrating keratoplasty, DSAEK = Descemet stripping automated endothelial keratoplasty, OCT = ocular coherence tomography, FA = fluorescein angiogram, ICG = indocyanine green, AMN = acute macular neuroretinopathy, OCTA = ocular coherence tomography angiography, PAMM = paracentral acute middle maculopathy, ICGA = indocyanine green angiography, VKH = Vogt-Koyanagi-Harada disease, MEWDS = multiple evanescence white dot syndrome, CRVO = central retinal vein occlusion, APD = afferent pupillary defect, CT = computed tomography, MRI = magnetic resonance imaging, Pt = patient, BID = twice daily, TID = thrice daily, QID = four times daily, qXh = every X hours, PO = oral, NR = not reported, IV = intravenous, CFT = central fovea thickness, VA = visual acuity, PF = prednisolone acetate, IVIG = intravenous immunoglobulin, LP = light perception, CF = counting fingers, HM = hand motions.

## Data Availability

Please refer to suggested Data Availability Statements in section “MDPI Research Data Policies” at https://www.mdpi.com/ethics (accessed on 30 December 2021).

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
