# Peer review of "Ocular Complications Following Vaccination for COVID-19: A One-Year Retrospective"

_vaccines, 2022, doi:10.3390/vaccines10020342_

Round 1

Reviewer 1 Report

Worthwhile addition to the literature.   Some care is needed to put all of these manifestations in context.  For example , what is the probability that these “side effects” are random occurrences.  This could be approached by discussing the association between date of vaccination and occurrence of symptoms.  

some discussion on leading recommendations on whether additional boosters are recommended in the setting of one of these side effects is recommended.  

When available please provide the delta of visual acuity.  For example in the context complication section you discuss average VA which is quite poor.   What was acuity prior to the complication.  

Author Response

Please see the appropriate section in the attached document.

Reviewer 2 Report

The authors described the occular manifestations of covid-19 vaccinations as the complication very properely and included all the significant and relevant reports.

The only remaining issues are; 

1-revision on the abstract that should be addressed on the name of the ocular complications

2 -on the discussion, that should be more coherantely and conclusion should be better in writting.

Author Response

Please see the appropriate section of the attached document. 

Reviewer 3 Report

Haseeb et al. report a comprehensive review of clinical ocular adverse events following COVID-19 vaccination. The authors raise the questions of a direct correlation of the adverse responses of these cases against various COVID-19 vaccines as possible side effects with potential serious post-immunization complications of COVID-19 vaccinations. The report is timely and would be very important. However, some of the images were not understandable for non-experts without assisting explanations. Also, the types of COVID-19 vaccines may be essential to consider the potential side effects, but some descriptions do not specify the types. Below are my specific comments.

  1. Figure 1. Please capitalize symbols in each panel.
  2. Lines 173, 187, 242, and 292. Please correct the double spaces.
  3. Figure 2. Please add assisting information to the panels, for example, what are the left and right images
  4. lines 264-265, the authors described “Fundus examination revealed serious retinal detachments”, but readers may not be able to find it. I suggest the authors add arrows or healthy eye images.
  5. Lines 271-273. This is not clear to this reviewer towards the conclusion that “there may be a common immunologic link between vaccination against and infection with COVID-19 which contributes to the development of VKH”. Please consider rewriting this section.
  6. Line 285 and 477. Please use consistent vaccine names. CoviShield vaccine or AZD1222.
  7. Line 294. Please specify if known the vaccine type.
  8. Line 373. What is the possible explanation of the vitamin D deficiency and the adverse effect?
  9. Figure 4. Please clarify “the teardrop-shaped macular lesion”, “hyperreflectivity of the outer nuclear and plexiform layers”, “disruption of the ellipsoid zone corresponding to the lesion”, and “subtle dropout in the deep capillary plexus corresponding to the lesion” in the images.
  10. Figure 5. The authors described “arrows show the areas of disruption and segmentation of the ellipsoid zone in the right eye and thinning of ellipsoid zone”. This is not clear to this reviewer from the images in Figure 5. Please add additional information.
  11. Lines 774-776. The authors proposed the cellular immune response against mRNA molecules as a possible mechanism of this adverse effect. It has been reported that COVID-19 mRNA vaccines have modified nucleotides to reduce their immunostimulatory potential (ex., ACS Cent. Sci. 2021, 7, 748-756). Please consider briefly describing this point for increasing clarity.

Author Response

(The authors gave the same response as above.)

Reviewer 4 Report

The manuscript of Haseeb et al. is a comprehensive review of the undesirable and adverse effects of COVID-19 vaccination in the eye. However, the manuscript is divided into sections like an original article.   In my opinion, authors should choose whether to provide a general overview of the topic, or whether to present original data as a result of a meta-analysis conducted on works in the literature.

Nevertheless, as a review, the manuscript is
exhaustive and detailed.  

Author Response

(The authors gave the same response as above.)

Round 2

Reviewer 4 Report

I still think that the manuscript's structure is dispersive. However, I recognize that the content has some aspects of originality and novelty

Author Response

We have revised the manuscript with a different framework. Much of the content of the review is the same, but it is presented more in the style of a review as opposed to an original investigative paper, as you have identified and suggested. The discussion section was eliminated and sections about mechanisms (when relevant) were added to the appropriate manifestation sections. We hope this is suitable for you and appreciate your feedback. 
